# Mapping the heritability of disease: a nationwide study

Janne Auning [1,2,3] ✉, Betina B. Trabjerg[1,3], Julie Werenberg Dreier [1,3], Bjarni Jóhann Vilhjálmsson [1,4,5,7] & Jakob Christensen [2,6,7]

Heritability estimates are essential for understanding genetic and environmental contributions to disease, yet large-scale studies remain scarce. In this study, we leverage the Danish national health registers, including medical records for more than 10 million individuals, to estimate heritability for more than 1000 health outcomes. We estimate heritability using both twins and siblings born in Denmark between 1955-2021, providing insight into the influence of shared sibling environment with estimates that show strong concordance with published twin studies and meta-analyses. We consider the impact of left-truncation by conducting analyses in both the full cohort and in individuals born after 1977. In a nested genotype case-cohort sample, we contrasted twin- and sibling-based heritabilities for psychiatric and neurological disorders with single-nucleotide polymorphism (SNP)-heritability, revealing disorder-specific "missing heritability" gaps. Together, these results map disease heritability in a single population, providing comprehensive insights for future genetic studies and preventive strategies using population health registers.

Accurately quantifying genetic contributions to disease is critical for developing risk prediction models and guiding precision medicine initiatives[1]. However, existing heritability estimates are often disease-specific, derived from selected cohorts, and may not generalise to broader populations[2,3]. To address this, large-scale and systematic approaches are needed to assess and correct for these potential biases.

Twin studies have long been a cornerstone of heritability research, offering a natural experiment for studying genetic and environmental influences[4,5]. By comparing phenotypic concordance between monozygotic (MZ) twins, who are genetically identical, and dizygotic (DZ) twins, who on average share 50% of their segregating genetic variation, these studies provide estimates of narrow-sense heritability. While this design mitigates confounding from shared environment, it also faces limitations.

Twin studies rely on dedicated twin cohorts that not only demand substantial resources to establish and maintain but may also suffer from recruitment biases, with male and DZ twin pairs being underrepresented[6,7]. Moreover, twins with specific characteristics or from higher socio-economic backgrounds may be more likely to participate in follow-up, exacerbating potential biases[7,8]. Even in registers with minimal selection, such as the Danish Twin Registry, information on zygosity is questionnaire-based, with zygosity being unknown for up to 25% of twins[9,10].

Beyond cohort limitations, heritability estimates themselves are sensitive to methodological and data-related biases. Data censoring as well as differences in reporting practices between countries are known to affect estimates[11]. Differences in estimation frameworks, whether based on twins, families, or single-nucleotide polymorphism (SNP) data, may also produce divergent results, with substantial gaps

[1]National Centre for Register-Based Research, Department of Public Health, Aarhus University, Aarhus, Denmark. [2]Department of Clinical Medicine, Aarhus University, Aarhus, Denmark. [3]Centre for Integrated Register-Based Research (CIRRAU), Aarhus University, Aarhus, Denmark. [4]Bioinformatics Research Centre, Aarhus University, Aarhus, Denmark. [5]Novo Nordisk Foundation Center for Genomics Mechanisms of Disease, Broad Institute of MIT and Harvard, Cambridge, MA, USA. [6]Department of Neurology, Aarhus University Hospital, Affiliated Member of Epi-CARE, Aarhus, Denmark. [7]These authors contributed equally: Bjarni Jóhann Vilhjálmsson, Jakob Christensen. ✉e-mail: jah@clin.au.dk

**Table 1 | Summary of twin and sibling cohorts**

|  | All pairs | Same-sex pairs | Opposite-sex pairs |
|---|---|---|---|
| Number of twin pairs (1977–2021) | 42,706 (50.9% males) | 27,326 (51.4% males) | 15,380 (50% males) |
| Number of twin pairs (1955–2021) | 56,994 (50.9% males) | 36,878 (51.4% males) | 20,116 (50% males) |
| Median follow-up time, years (IQR) (1977–2021) | 18.2 (9.9–27.1) | 18.5 (9.7–27.8) | 17.8 (10.1–26.9) |
| Median follow-up time, years (IQR) (1955–2021) | 23.3 (12.5–40.8) | 24.5 (12.8–43.6) | 22.4 (12.5–40.0) |
| Number of sibling pairs[a] (1977–2021) | 716,344 (51.4% males) | 27,326 (51.4% males) | 689,018 (51.4% males)[b] |
| Number of sibling pairs[a] (1955–2021) | 1,115,264 (51.5% males) | 36,878 (51.4% males) | 1,078,386 (51.6% males)[b] |
| Median age difference within pairs, years (IQR) (1977–2021) | 2.40 (1.80–3.00) | 0 | 2.40 (1.80–3.00) |
| Median age difference within pairs, years (IQR) (1955–2021) | 2.40 (1.70–3.00) | 0 | 2.40 (1.80–3.00) |
| Median follow-up time, years (IQR) (1977–2021) | 20.4 (10.6–30.0) | 18.5 (9.70–27.8) | 20.4 (10.6–30.1) |
| Median follow-up time, years (IQR) (1955–1977) | 29.3 (15.3–45.0) | 24.5 (12.8–43.6) | 29.5 (15.5–45.0) |

[a]Including twins.
[b]Including both same-sex and opposite-sex sibling pairs along with the opposite-sex twin pairs.

between twin/family-based and SNP-based estimates, a discrepancy often referred to as the "missing heritability" problem[12]. Each approach is subject to distinct assumptions and artefacts, complicating cross-study interpretation and limiting the utility of these estimates for translational research. Large-scale meta-analyses such as the Meta-Analysis of Twin Correlations and Heritability (MaTCH) have sought to provide systematic overviews of heritability across traits by aggregating results from thousands of classical twin studies[3]. While these efforts have been invaluable as reference resources, they necessarily combine heterogeneous cohorts with differing recruitment strategies, diagnostic practices, and follow-up durations, which can introduce variability and limit comparability across disease domains and populations.

A recently proposed method presented by Lakhani et al. (CaTCH) seeks to overcome these biases while retaining the advantages of twin-based heritability estimation[2]. Instead of relying on zygosity, this method stratifies twin pairs into same-sex and opposite-sex groups, using this characteristic as a robust proxy to differentiate genetic similarity. Because sex is recorded reliably in health and insurance registers, this approach enables heritability estimation at population scale using electronic health data while reducing information and selection biases. Furthermore, the framework can be expanded to include non-twin siblings, potentially increasing statistical power and enabling the analysis of rarer outcomes. This also allows for the contrasting of twin and sibling estimates, which could identify environmental influences shared between siblings in specific disease outcomes.

However, the reliance on insurance-based data in the CaTCH study introduces several well-recognised limitations, including ascertainment bias, limited follow-up, and incomplete phenotype capture[13–16]. In contrast, the Danish national health registries provide nationwide, population-based coverage with virtually complete follow-up and systematic recording of all hospital contacts, thereby minimising selection and ascertainment biases. In this study, we therefore apply and extend the novel CaTCH framework to these registries, incorporating both twin and sibling data to estimate heritability for more than one thousand disease outcomes. In addition, we assess how estimates derived from twins and siblings align with SNP-based methods, providing a systematic evaluation of methodological effects across a wide disease spectrum. The resulting heritability map offers a comprehensive resource for refining disease risk models, informing screening strategies, and guiding precision prevention efforts.

## Results

In all, 56,994 (50.9% males) twin pairs and 1,115,264 (51.54% males) sibling pairs were identified from the Danish birth registers and subsequently linked to their individual diagnostic information (Table 1).

Among these, 42,706 (50.9% males) twin pairs and 716,344 (51.4% males) sibling pairs were included in the 1977–2021 cohorts. The median follow-up time was increased by 5.1 and 8.9 years in the extended 1955 compared to the 1977 birth cohort for the twin and sibling cohorts, respectively. The median age difference within sibling pairs was consistently 2.40 years across birth cohorts. The probability of a same-sex twin pair being monozygotic was 0.437 in the 1977 cohorts and 0.456 in the 1955 cohorts.

### Mapping heritability using twins

As a first step in constructing our nationwide heritability map, we applied the CaTCH method to our register-based twin data[2]. A total of 385 diseases and disorders had at least 400 cases in the 1955 birth cohort, 186 of which resulted in non-zero heritabilities (Fig. 1a). Of these, 85 phenotypes were categorised as early-onset and additionally estimated in the 1977 birth cohort (Fig. 1b), where individuals were followed from birth. However, estimates from the 1977 and 1955 birth cohorts showed high concordance, suggesting minimal effects from left truncation of diagnostic information in the 1955 birth cohort (Fig. 1c). The few phenotypes with larger discrepancies showed substantial differences in the sex distribution of cases between cohorts, with a sex ratio of 3.76 in the 1977 cohort compared to 3.02 in the 1955 cohort for cholelithiasis, and 1.23 compared to 2.10 for foetal distress (Supplementary Fig. 1). Furthermore, the larger sample size of the 1955 cohort increased statistical power, resulting in narrower confidence intervals. As illustrated in Fig. 1d, this is reflected in the inverse-variance weighted mean heritability and corresponding 95% confidence intervals for early- and late-onset phenotypes, estimated separately in the 1955 (purple) and 1977 (green) cohorts.

Overall, the inverse-variance weighted mean heritability of the 1955 estimates was 0.445 (95% CI: 0.429, 0.462). Clustering the estimates in functional domains, Fig. 1a, b shows the distribution of our estimates across these groups. We found the largest contribution from additive genetic variance in the endocrine/metabolic ($h^2 = 0.764$, 95% CI: 0.687, 0.841), genitourinary ($h^2 = 0.744$, 95% CI: 0.639, 0.849), digestive system ($h^2 = 0.583$, 95% CI: 0.518, 0.648), and circulatory system ($h^2 = 0.583$, 95% CI: 0.445, 0.722) disorders (Supplementary Fig. 2a). The lowest estimates were among the domains of injuries and poisonings ($h^2 = 0.273$, 95% CI: 0.237, 0.310) and infectious diseases ($h^2 = 0.256$, 95% CI: 0.193, 0.319).

### Expanded mapping from the sibling cohort

Building on the twin-based framework, we expanded our analysis to include full sibling pairs. The relatively small size of the twin cohort resulted in high uncertainty in many estimates, particularly for less common phenotypes. Incorporating siblings substantially increased the number of analysable cases, enabling more precise heritability estimates and extending coverage to a broader range of disorders.

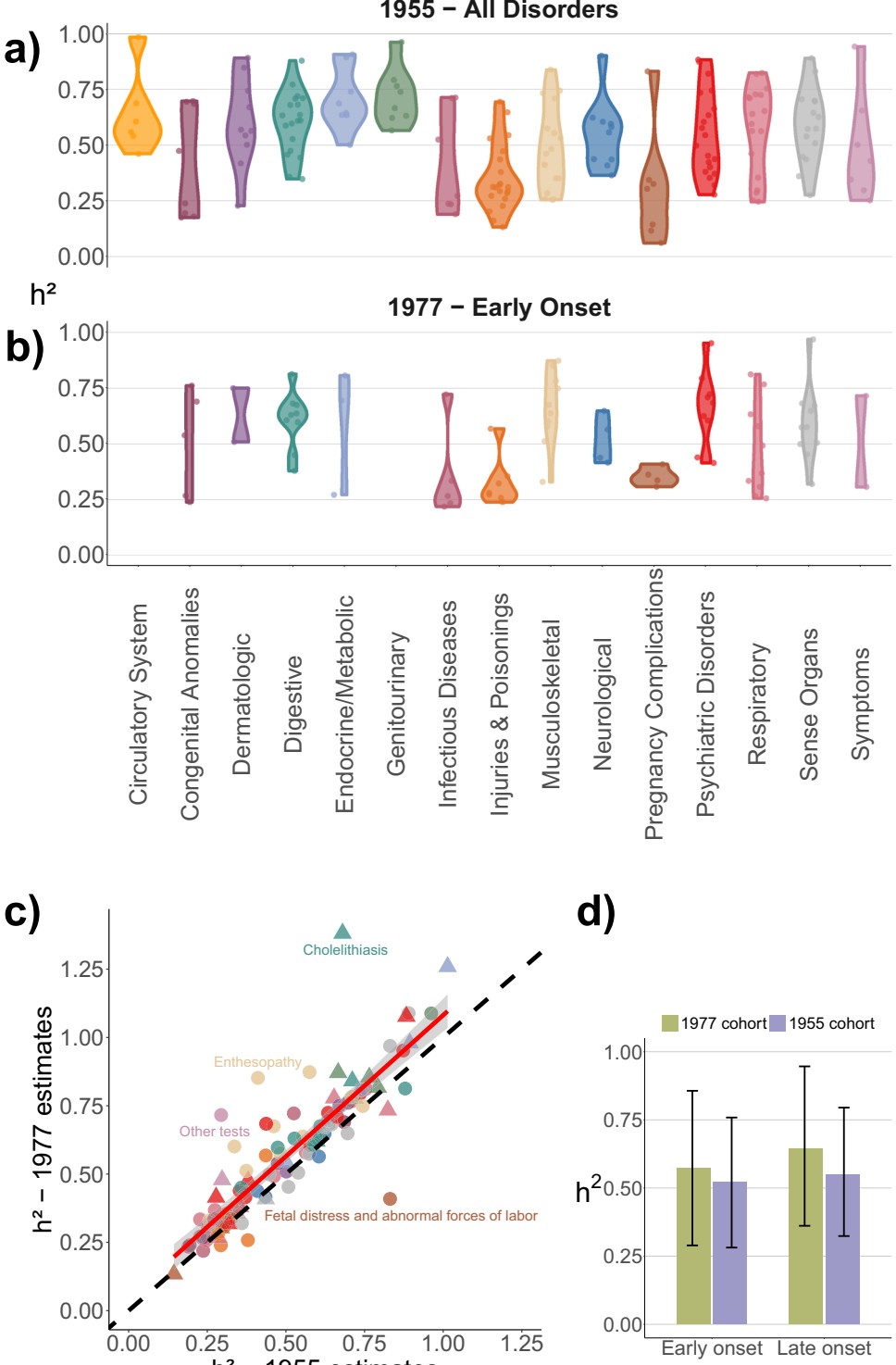

**Fig. 1 | Twin heritability estimates across functional domains and birth cohorts.** Distribution of liability-scale heritability estimates across functional domains, stratified by onset classification. **a** includes all phenotypes from the 1955 birth cohort, irrespective of onset classification, while (**b**) presents early-onset phenotypes estimated from the 1977 birth cohort. **c** Comparison of heritability estimates for phenotypes in the 1955 and 1977 birth cohorts. Each point represents a phenotype, coloured according to its functional domain and shaped by onset classification (circle = early onset; triangle = late onset). The red line indicates the linear regression fit with standard error, while the dashed black line represents the identity line (slope = 1). **d** Inverse-variance weighted (IVW) heritability estimates for early- (*n* = 85) and late-onset (*n* = 27) phenotypes in the 1955 and 1977 birth cohorts. Each bar represents a single IVW mean based on the subset of phenotypes where estimates could be obtained from both birth cohorts. Error bars denote 95% confidence intervals.

The 1,115,264 sibling pairs resulted in a threefold increase in coverage of analysable phecodes (Table 2). We found 1139 diseases and disorders with a minimum of 0.5% cases and without sex imbalances, 593 resulting in non-zero heritabilities, and 321 were categorised as early-onset diseases (Supplementary Fig. 3). To compare these results

to those of our twin cohorts, we excluded phenotypes from the sibling results that displayed sex ratios substantially different from those in the twin cohorts. This was done to ensure comparability, as sex imbalance biases the results of the CaTCH method. When doing this, our twin and sibling results were generally concordant across both the 1955 and 1977 cohorts (Fig. 2). However, sibling-based estimates were consistently larger than those from twins, with the inverse-variance weighted mean heritability across all phenotypes in the sibling cohort being 0.514 (95% CI: 0.509, 0.519). Yet, the overall pattern across functional domains remained the same with the largest contribution from additive genetic variance in the endocrine/metabolic ($h^2 = 0.739$, 95% CI: 0.718, 0.760) and genitourinary ($h^2 = 0.717$, 95% CI: 0.679, 0.755) domains and the lowest estimates were among the domains of injuries and poisonings ($h^2 = 0.360$, 95% CI: 0.345, 0.375) (Supplementary Fig. 2b).

**Table 2 | Phecode counts across cohorts**

|  | 1977 Birth Cohort | 1955 Birth Cohort |
|---|---|---|
| Twin Cohort | 256 | 385 |
| Sibling Cohort | 897 | 1139 |

Number of phecodes with minimally 0.5% cases in the different cohorts. These were subsequently filtered to include only estimates with FDR < 0.05. The 1977 results were further filtered to include only phenotypes that could be categorised as early onset.

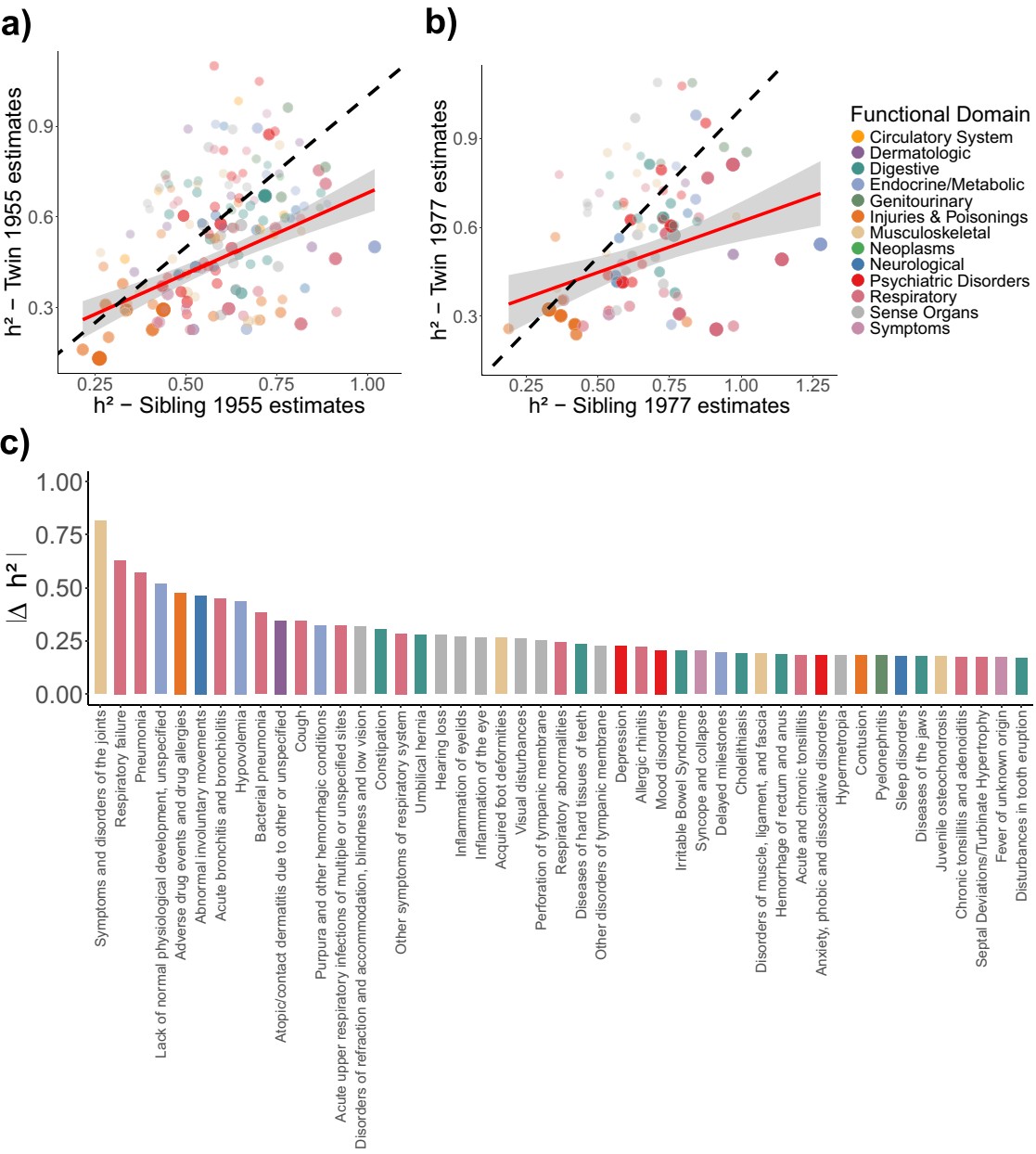

**Fig. 2 | Comparing sibling and twin estimates.** Comparison of liability-scale heritability estimates derived from sibling pairs (x-axis) with those from twin pairs (y-axis) using **a** the 1955 cohort and **b** the 1977 cohort. Each point represents a phenotype, coloured according to its functional domain and sized by weight. The red line indicates the weighted linear regression fit with standard error, while the dashed black line represents the identity line (slope = 1). **c** 40 largest absolute differences in heritability estimates between sibling and twin pairs, ranked in descending order. Bar colour corresponds to the functional domain of the phenotype.

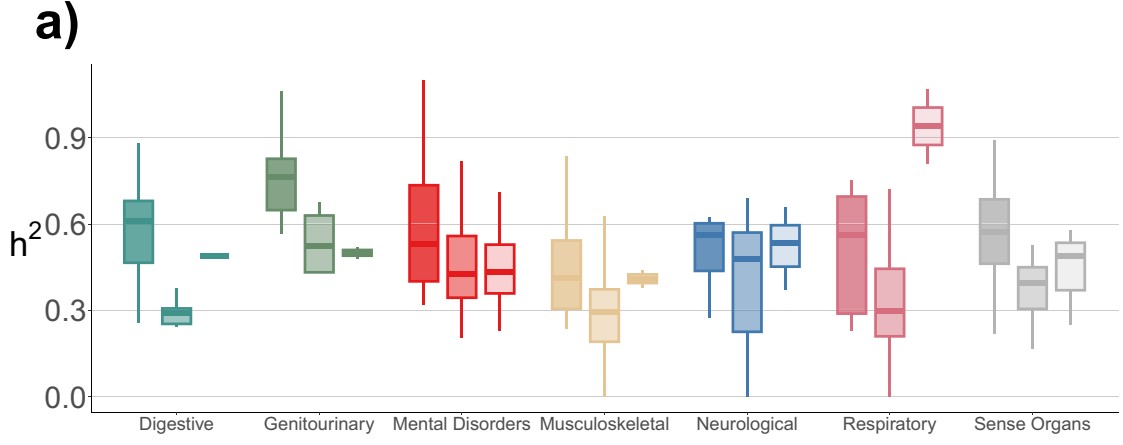

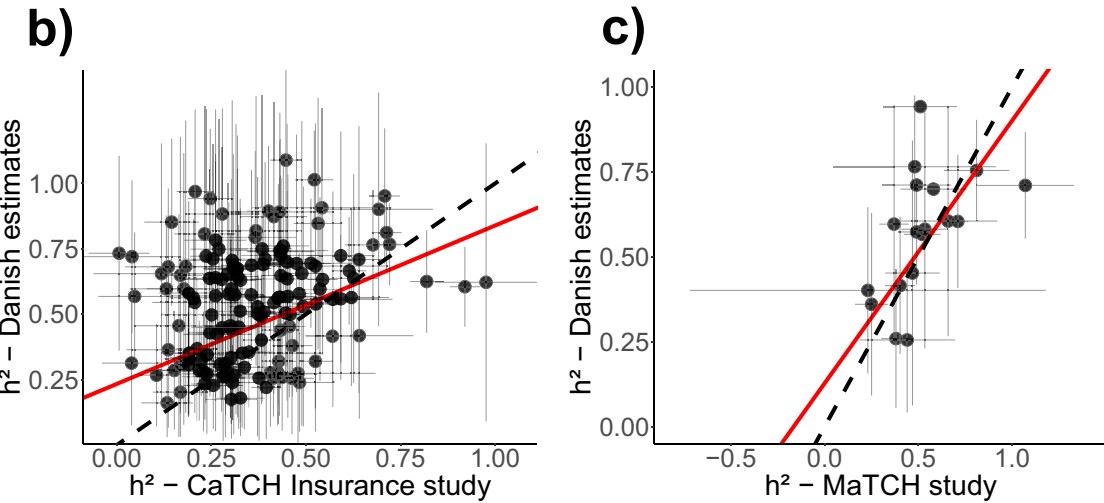

**Fig. 3 | Comparison of Danish twin heritability estimates with previous twin-based studies. a** Distribution of heritability estimates across functional domains for the Danish dataset (left; $n = 128$), the CaTCH insurance study[2] (middle; $n = 96$), and the MaTCH meta-analysis[3] (right; $n = 17$). Boxplots show the median (centre line) and interquartile range (box; 25th–75th percentiles); whiskers extend to the most extreme values within $1.5 \times$ IQR. Outliers are not shown. **b** Weighted regression comparing Danish heritability estimates to those from the CaTCH study ($n = 148$). (c) Weighted regression comparing Danish estimates with those from the MaTCH meta-analysis ($n = 18$). In (**b** + **c**) each point represents a phenotype with non-zero heritability in both datasets with 95% confidence intervals. The red line indicates the weighted linear regression fit, while the dashed black line represents the identity line (slope = 1).

The phenotypes that deviated the most from their twin counterpart were predominately in the respiratory ($h^2_{\text{diff}} = 0.276$) or endocrine/metabolic ($h^2_{\text{diff}} = 0.262$) functional domains (Fig. 2c). Notably, disorders such as respiratory failure, pneumonia, and bacterial pneumonia showed some of the largest absolute differences in heritability estimates. Conversely, phenotypes in the injuries and poisonings ($h^2_{\text{diff}} = 0.138$), genitourinary ($h^2_{\text{diff}} = 0.124$), and psychiatric ($h^2_{\text{diff}} = 0.111$) functional domains had the smallest mean difference in $h^2$ between twin and sibling estimates.

As with the twin results, the heritability estimates derived from siblings were highly consistent between the two birth cohorts when comparing phenotypes with similar sex ratios (Supplementary Fig. 4).

**Benchmarking against existing twin studies**
To evaluate our findings, we examined the degree of concordance with prior literature by comparing our twin heritability estimates to the original CaTCH[2] estimates as well as those of the Meta-Analysis of Twin Correlations and Heritability (MaTCH)[3] (Fig. 3).

Our results were largely consistent with those of the MaTCH study (Fig. 3a, c), but our mean twin heritability of 0.445 (95% CI: 0.429, 0.462) was smaller than their mean of $h^2 = 0.593$ (95% CI: 0.577, 0.608).

Yet, compared to the CaTCH results, our results show greater concordance with the traditional findings, as CaTCH reports a mean of 0.315 (95% CI: 0.296, 0.334). Particularly, the neurological, psychiatric, and musculoskeletal disorders displayed high concordance with the MaTCH estimates. Phenotypes in the digestive, respiratory, and sense organ categories were further away. However, they were still closer to the MaTCH distribution than the CaTCH estimates were.

**Investigating the missing heritability gap**
To further investigate the genetic architecture underlying a range of complex disorders in the Danish population, we examined the effect of common variation and compared it to the total additive genetic contribution, which our narrow-sense heritabilities quantify (Fig. 4).

To do this, we estimated $h^2_{SNP}$ in the iPSYCH cohort for 9 disorders and found that, on average, common variation accounted for 19.2% of phenotypic variance with 39.6% of additive variance for the neurological disorders and the 11.4% of phenotypic variance with 27.2% of additive variance for the psychiatric disorders.

Across most disorders, SNP-heritabilities were markedly lower than their twin- and sibling-based counterparts. This gap was especially pronounced for psychiatric disorders such as schizophrenia spectrum

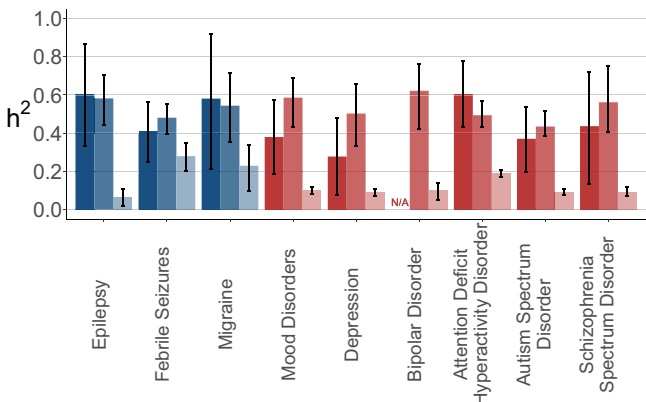

**Fig. 4 | Narrow-sense and SNP-heritability estimates for selected neurological and psychiatric disorders.** Heritability estimates for a range of neurological (blue) and psychiatric (red) disorders comparing different methods: twin-based (dark, left; n = 113,980), sibling-based (medium, middle; n = 2,230,520), and SNP-based (lightest, right; n: See Supplementary Data 3). Estimates are shown for the 1955 birth cohort. Error bars indicate 95% confidence intervals of the heritability estimates.

disorder, ASD, and BD, as well as the neurological condition epilepsy, where SNP-heritability accounted for only a fraction of the total additive genetic signal. In contrast, phenotypes like migraine and febrile seizures showed a smaller gap, suggesting a larger proportion of their genetic architecture may be attributable to common variants.

To further characterise these patterns, we examined the estimated polygenicity ($p$) and selection coefficients ($\alpha$) for each disorder (Supplementary Fig. 7). Psychiatric disorders exhibited higher polygenicity and more negative selection coefficients than neurological disorders, indicating that their genetic architecture is shaped by a larger number of small-effect variants subject to stronger purifying selection.

## Discussion

Understanding the genetic and environmental foundations of disease in a population is crucial to advancing disease risk prediction, yet traditional studies have been limited in scalability and applicability across diseases. By leveraging a novel approach in the Danish electronic health records, we address these limitations, creating the most comprehensive mapping of genetic contributions to disease in the Danish population to date with estimates of heritability for more than 1000 phenotypes.

To do this, we created two birth cohorts for both the twin and the combined twin and sibling samples, which enabled heritability estimation across both early- and late-onset phenotypes. Left truncation of diagnostic data has previously been shown to bias prevalence and survival estimates, particularly for disorders with variable onset ages or chronic trajectories[17–19]. However, we observed high concordance between estimates from the 1955 and 1977 cohorts, suggesting that left truncation has a surprisingly small impact on heritability estimation (when restricting to phecodes with sufficient number of cases to allow for heritability estimation). This suggests that we can use the 1955 birth cohort for most phenotypes, regardless of onset, which had a larger sample size, improving accuracy and coverage. A few phenotypes showed larger discrepancies between cohorts, which were primarily associated with pronounced differences in the sex distribution of cases. For instance, the sex ratio for cholelithiasis differed substantially between cohorts, as did the ratio for foetal distress. The latter likely being due to the earlier records capturing maternal rather than infant diagnoses, leading to a sex imbalance that can bias estimates produced by the CaTCH method.

The heritability estimates from this 1955 birth cohort varied widely across disease domains. The disease domains with the highest

heritability estimates were genitourinary, endocrine/metabolic, and circulatory system disorders, while the lowest estimates were found for injuries, poisonings, and infections. These domain-level patterns are broadly consistent with findings from both the CaTCH[2] and MaTCH[3] studies, which also reported elevated heritabilities for neuropsychiatric and endocrine conditions, and considerably lower estimates for external injuries and obstetric traits. Notably, psychiatric disorders such as ADHD, autism, and schizophrenia spectrum disorders consistently ranked among the most heritable traits across studies, supporting the strong and replicable genetic component of these phenotypes[2,3].

Further comparisons of our twin heritability estimates with the twin estimates from CaTCH and MaTCH studies revealed that while our results were generally larger than those reported in CaTCH, they were closely aligned with those in MaTCH, particularly for neurological, psychiatric, and musculoskeletal disorders. This difference to CaTCH estimates likely reflects ascertainment biases inherent to the insurance claims data used in the CaTCH study, such as incomplete capture of diagnoses and underrepresentation of individuals with limited healthcare access. Our use of nationwide register data, with longer follow-up and improved ascertainment, allows for more robust phenotype capture. The improved resolution in our musculoskeletal and endocrine estimates, for instance, may be due to better chronic disease documentation over time in the Danish National Patient Register.

While these twin-based results were highly promising, the reliance on large case numbers for the CaTCH method limited the applicability of studying rare diseases. We sought to counter this limitation by extending the framework to include non-twin siblings, thereby greatly increasing both coverage and statistical precision. As expected from the design, sibling-based estimates were consistently larger than those from twins. This inflation is attributable to greater environmental variance between siblings than twins, which reduces the shared environmental component and can lead to an overestimation of the genetic contribution[20]. Importantly, this discrepancy can also serve as a signal; phenotypes with major differences between twin and sibling estimates may be more influenced by shared environmental factors that twins, but not siblings, experience similarly. Such factors could include prenatal or timing-specific exposures that are especially detrimental during critical developmental windows. We observed the largest deviations in the endocrine/metabolic and respiratory domains, particularly for phenotypes such as respiratory failure and pneumonia, which are likely to be shaped by early-life exposures such as infections or household conditions. In contrast, domains such as psychiatric, musculoskeletal, and injuries showed minimal differences, suggesting that heritability estimates in these categories are more robust to the choice of family design.

Leveraging our results, we further explored the genetic architecture of a range of complex brain disorders by comparing our estimates for narrow-sense heritabilities in the full Danish population with SNP-heritabilities. Using the Danish iPSYCH sample, we found that common genetic variants accounted for only a fraction of the total additive genetic variance, with the largest discrepancies observed in schizophrenia spectrum disorder and epilepsy, consistent with previous findings[12,21,22]. Notably, febrile seizures exhibited a much smaller gap compared to epilepsy. This is unexpected given the clinical and genetic overlap between febrile seizures and epilepsy[22,23], suggesting that common variants may more strongly drive the genetic architecture of febrile seizures, whereas epilepsy may have a more considerable contribution from rare or structural variants that are possibly subject to deleterious selection[24].

Altogether, this framework offers a scalable approach to estimating heritability across a wide range of diseases and investigating the genetic architecture of disease. However, the method does not apply to sex-specific phenotypes which restricts our ability to assess the heritability of conditions that predominantly or exclusively affect

one sex, such as placenta previa or prostate cancer, or phecodes that are groupings of sex-specific subgroups, such as Congenital anomalies of genital organs that contain the subgroups Congenital anomalies of female genital organs and Congenital anomalies of male genital organs. To address this, future work could incorporate traditional zygosity-based estimates from the Danish Twin Registry. In the present study, we chose to focus on twins and siblings rather than full pedigrees to reduce the risk of shared environmental factors confounding the heritability estimates, but the framework could also be expanded with pedigree-based methods, for which the Danish registries are ideal[16,25]. Incorporating these methods would also allow for further validation of our results. Such a comparison would help assess the accuracy of using sex as a proxy for genetic similarity and strengthen confidence in applying this approach to other populations.

Furthermore, we did not account for age differences within sibling pairs. Larger age gaps may reduce shared environmental exposure and affect within-family phenotypic correlations, which could partly contribute to the observed differences between sibling- and twin-based heritability estimates. Although the average age difference in our cohort was small (2.4 years), future work could examine this more directly by adjusting for age in the linear mixed model.

In summary, we used a novel method to develop a comprehensive heritability map of more than one thousand phenotypes. This resource provides a foundation for integrating heritability estimates into clinical and public health applications, such as refining disease risk prediction models, informing screening strategies, and identifying individuals at heightened genetic risk for targeted interventions. By improving our understanding of genetic contributions to disease, this work supports the development of precision medicine approaches and enhances early detection efforts, ultimately contributing to improved patient outcomes and more efficient healthcare resource allocation.

## Methods

### Narrow-sense heritability estimated using twin and sibling data

**Study design and population.** The first part of the study was undertaken as a register-based analysis of all twins and sibling pairs born in Denmark between the 1st of January 1955 and the 31st of December 2021. The cohorts were created using the Danish Civil Registration System (CRS)[26], where twin and full sibling pairs were identified and linked through the matching of the unique personal identification numbers of their parents, thus creating two cohorts: a twin cohort and a sibling cohort, the latter containing both twin and non-twin sibling pairs.

The twins were classified as such, granted that these pairs had a date of birth within 1 day of each other as well as identical places of birth. Only the first twin pair, born to a mother and father after 1955, was included. Each non-twin sibling pair was identified as children born to the same mother and father between 11 months and 4 years apart. If a parent pair had had more than two children, only the sibling pair born closest together were included in the cohort to minimise sibling pair environmental variance. All persons were followed until either death or emigration, the death or emigration of their sibling, or the 31st of December 2021.

Each individual included in these two cohorts was linked with their individual diagnostic information from the Danish National Patient Register (NPR)[27], which includes data on hospital discharges from 1st January 1977, with outpatient information incorporated since 1st January 1995. Diagnostic details are based on the International Classification of Diseases (ICD), using the 8th revision (ICD-8) between 1977 and 1993 and subsequently transitioning to the 10th revision (ICD-10) from 1994 to 2021. Since we only have diagnostic data from 1977 and onwards, we created two versions of both cohorts. The first and main version included all twin and non-twin siblings born between 1955 and 2021 and who were alive on the 1st of January 1977. This was

done to enable the study of disorders with onset in late life and to increase the sample size. The second version of both cohorts included all twin and non-twin siblings born between 1977 and 2021 to investigate any effect of left truncation on the heritability estimates. Phenotypes were classified as early-onset if they had a cumulative incidence of at least 10% among individuals aged 0–18 years[28].

The ICD codes were subsequently translated to phecodes, a phenotyping system that organises ICD codes into broader categories of clinical relevance[29,30]. Since the study spans ICD-8 and ICD-10 coding periods, diagnoses that were recorded using ICD-8 classification were first converted to their equivalent ICD-10 codes, using a conversion mapping of ICD-8 to ICD-10[31], as only ICD-9 and ICD-10 maps exist to convert ICD codes to phecodes. We made a few modifications to the original phecode map, updating the epilepsy phecode definition to follow[32] and adapting the psychiatric phecodes according to[33].

For analyses, we included all phecode categories with a disease prevalence in the resulting cohorts larger than 0.5% and at least five pairs of concordant same-sex and opposite-sex twins to allow for robust heritability estimation, while complying with data protection guidelines[2] (Supplementary Data 1). Similarly, phecode categories with a female-to-male or male-to-female ratio of more than 5 were filtered away (Table 2) along with their parental phecodes to avoid categories that appear to have a balanced sex distribution, but are in fact a mixture of two or more imbalanced phecodes (e.g., Congenital anomalies of female genital organs (751.11), Congenital anomalies of male genital organs (751.12) which aggregate in the category Congenital anomalies of genital organs (751.1))[2]. We estimated the heritability of the resulting disorders in both the twin and the sibling cohorts.

**Estimating heritability from a modified Falconer's formula.** With the Falconer's formula, the heritability of a trait can be estimated as follows[4]:

$$h^2 = 2(r_{MZ} - r_{DZ}) \tag{1}$$

where $r_{MZ}$ and $r_{DZ}$ are the correlations of monozygotic and dizygotic twins, respectively. Yet, when twin zygosity is not known, a modified formula can be utilised as presented in[2]:

$$h^2 = \frac{2}{p}(r_{SS} - r_{OS}) \tag{2}$$

where $r_{SS}$ and $r_{OS}$ are the same-sex and opposite-sex correlations respectively, and $p$ is the probability of a same-sex pair being monozygotic. Consequently, twin pairs were stratified into same-sex and opposite-sex twins, where opposite-sex pairs will necessarily be dizygotic and same-sex pairs will be a mixture of monozygotic and dizygotic twins. Assuming Weinberg's rule, the probability of a pair of dizygotic twins being same-sex is 50%[34], and from this, the probability ($p$) of a twin pair being monozygotic, given they are the same sex, can be estimated as follows:

$$p = p(MZ|SS) = \frac{p(MZ)}{p(SS)} \tag{3}$$

here, $p(MZ|SS)$ denotes the probability that a twin pair is monozygotic, given that they are same-sex. This is calculated as the ratio of the overall probability of being monozygotic, $p(MZ)$ to the probability of being a same-sex pair, $p(SS)$.

The overall probability of being monozygotic, $p(MZ)$, can be expressed in terms of the proportion of opposite-sex pairs:

$$p(MZ) = 1 - 2p(OS) = 1 - 2\frac{N_{OS}}{N_{all}} \tag{4}$$

here, $p(OS)$ is the probability of being an opposite-sex pair, with $N_{OS}$ being the number of opposite-sex pairs and $N_{all}$ being the total number of pairs.

Finally, the probability of being a same-sex pair is given by:

$$p(SS) = \frac{N_{SS}}{N_{all}} \qquad (5)$$

where $N_{SS}$ is the number of same-sex pairs.

In the analyses with the full sibling cohort, all non-twin sibling pairs in the cohort, irrespective of sex composition, were categorised as opposite-sex pairs, reflecting how full siblings are as genetically similar as dizygotic twins.

**Variance component model.** Assuming that $r_{OS}$ is a good proxy for $r_{DZ}$ and that $r_{SS}$ is a mixture of $r_{DZ}$ and $r_{MZ}$, the phenotypic correlation within each of the two groups of twins are computed by fitting the following linear mixed model in R:

$$y = X\beta + u_{pair} + u_{onlySS} + e \qquad (6)$$

where $y = 1$ for individuals with the specific disorder and $y = 0$ for individuals without it, $X\beta$ are the fixed effects sex and age, while the random effect $u_{pair}$ is common in any pair and the random effect $u_{onlySS}$ is common only in pairs of same-sex twins. $e$ is the error term.

As $V_y = V_{pair} + V_{onlySS} + V_e$, the covariance between an opposite-sex pair is $V_{pair}$ and $V_{pair} + V_{onlySS}$ for a same-sex pair. This results in the following variance components:

$$V_{SS} = V_{pair} + V_{onlySS} \qquad (7)$$

$$V_{OS} = V_{pair} \qquad (8)$$

$$V_{total} = V_{pair} + V_{onlySS} + V_e \qquad (9)$$

These variant components were then used to estimate the twin correlations on the observed scale:

$$r_{SS} = \frac{V_{SS}}{V_{total}} \qquad (10)$$

$$r_{OS} = \frac{V_{OS}}{V_{total}} \qquad (11)$$

**From observed to liability scale.** As all phenotypes were binary, the correlations were converted to liability scale as follows:

$$r_{SS\_liab} = \frac{(T - T_{SS})\sqrt{1 - \left(T^2 - T_{SS}^2\right)\left(1 - \frac{T}{i}\right)}}{i + T_{SS}^2(i - T)} \qquad (12)$$

Where:

$$T = \phi^{-1}(1 - K) \qquad (13)$$

$$i = \frac{\phi(T)}{K} \qquad (14)$$

$$T_{SS} = \phi^{-1}\left(1 - K + \frac{V_{SS}}{K}\right) \qquad (15)$$

$K$ is the prevalence of the phenotype in the cohort. The same procedure was used for $r_{OS}$.

**Bootstrap and adjustment for multiple testing.** 95% confidence intervals were computed using 500 bootstrap samples with the boot R package[35,36]. *P*-values were computed using a two-tailed z-test and corrected with the Benjamini–Hochberg FDR-correction to account for multiple testing[37]. Phenotypes with FDR < 5% were reported as significant.

**Benchmarking against published twin-based heritability estimates.** We extracted published twin-based heritability estimates for relevant diseases from peer-reviewed meta-analyses and primary studies, including those by Polderman et al. (MaTCH)[3] and Lakhani et al. (CaTCH)[2]. We mapped phecodes of our twin estimates to the corresponding phecodes of the CaTCH study. Mapping to phenotype definitions of the MaTCH study was done manually.

### SNP-heritability using genotype data

**The iPSYCH case-cohort sample.** In the second part of the study, the SNP-heritability, $h_{SNP}^2$, estimates were computed using data from the Lundbeck Foundation Initiative for Integrative Psychiatric Research (iPSYCH) case-cohort sample, a dataset of genotyped individuals sampled from the entire Danish population born between 1981 and 2008 with DNA samples collected from neonatal dried bloodspots[38,39]. Of the 1,657,449 individuals in the birth cohort, a total of 92,765 cases were genotyped, all individuals diagnosed with a major psychiatric disorder. Additionally, 42,912 individuals randomly sampled from the same birth cohort were genotyped as part of a random subcohort sample. The genotype data were imputed following the RICOPILI pipeline[40] and using the Haplotype Reference Consortium (HRC) as the reference panel[41].

The genotype data can be linked to diagnostic information based on ICD codes in the NPR[27], resulting in the conversion into phecode-defined disorders in persons with and without psychiatric disorders as described above.

**Estimating SNP-heritability from summary statistics.** Logistic regression genome-wide association studies (GWAS) were conducted for the following phecodes, for which phenotype information was available in the iPSYCH sample: epilepsy, febrile seizures, migraine, mood disorders, depression, bipolar disorder (BD), attention-deficit/hyperactivity disorder (ADHD), autism spectrum disorder (ASD), and schizophrenia spectrum disorder.

Prior to the association analyses, any samples with a KING-relatedness robust coefficient corresponding to 3rd-degree relationships or closer were filtered away[42]. Moreover, using principal component analysis (PCA), 20 principal components were computed to identify a genetically homogenous population, and any samples having more than 4.5 log distance units to the multidimensional centre of the 20 PCs were filtered away[43]. Finally, any variant with a minor allele frequency (MAF) smaller than 0.01 was excluded.

When conducting the logistic regressions, the first 20 PCs, sex, and age were included as covariates. The resulting summary statistics were used to compute the $h_{SNP}^2$, polygenicity ($p$) and selection coefficient ($\alpha$) using the Bayesian framework LDpred2-auto[44,45].

### Reporting summary

Further information on research design is available in the Nature Portfolio Reporting Summary linked to this article.

## Data availability

Data are not publicly available due to Danish data protection regulations. The study used individual-level data from the Danish Civil Registration System, the Danish National Patient Register, and the iPSYCH case-cohort sample, which are protected under national legislation. Data can only be accessed through secure servers and international researchers need a collaboration with a Danish research institution.

Maps to translate diagnosis codes can be found at ref. 31 (ICD8 to ICD10) and https://phewascatalog.org/phewas/#phe12 (ICD10 to phecodes).

## Code availability

Code used to map ICD10 diagnosis codes to phecodes and to generate results can be found at https://github.com/janneah/heritability, https://doi.org/10.5281/zenodo.18480233.

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

## Acknowledgements

J.A. was supported by The Central Denmark Region, the Novo Nordisk Foundation (NNF16OC0019126 and NNF22OC0075033), and a Lundbeck Foundation Fellowship (R335-2019-2339). J.C. and B.B.T. were supported by Novo Nordisk Foundation (NNF16OC0019126 and NNF22OC0075033), the Lundbeck Foundation (R402-2022–1485), the Central Denmark Region, and the Danish Epilepsy Association. J.W.D. was supported by The Independent Research Fund Denmark (4253-00007B, 3166-00134B, 4308-00142B). B.J.W. was supported by Independent Research Fund (2034-00241B), Lundbeck Fellow Grant (R335-2019-2339), and Danish National Research Foundation (P4).

## Author contributions

J.A. performed the data processing and analyses and wrote the manuscript. B.B.T. reviewed the code. J.W.D., J.C., and B.J.V. supervised the study. All authors contributed with revisions to the final manuscript.

## Competing interests

J.C. has received honoraria from serving on the scientific advisory board of UCB Nordic and Eisai AB, received honoraria for giving lectures from UCB Nordic and Eisai AB, and received funding for a trip funded by UCB Nordic. The remaining authors declare no competing interests.
