## [Transparent Peer Review file · Nature Communications]

Mapping the Heritability of Disease: A Nationwide Study

Corresponding Author: Ms Janne Auning

Version 0:

Reviewer comments:

Reviewer #1

(Remarks to the Author)

The study is using information about same-sex and distinct-sex siblings plus share or not shared date of birth of siblings as a proxy to monozygotic and dizygotic twin analysis.

The study is valuable and seems to agree well with previously published data.

This reviewer has the following questions (ideally answered in the revised manuscript).

1. Since mother and father information is available, why the authors did not consider analyzing heritability based on whole family rather on siblings?
2. Did the authors consider the issues of statistical efficiency? The equations they used were developed in pre-computer era for ease of manual computation.
3. There were a number of published studies regarding estimating heritability from administrative records the authors seem to be unaware of. Would the author consider augmenting their comparisons to prior estimates?

(Remarks on code availability)

Reviewer #2

(Remarks to the Author)

To estimate narrow-sense heritability, the authors identified all twin and sib-pairs born in Denmark between 1955 and 2021 using the Danish Civil Registration System, and created a twin cohort and a sibling cohort that is inclusive of the twin cohort. The authors performed data linkage to the Danish National Patient Register was done for both cohorts to identify disease diagnostic records. The authors then estimated narrow-sense heritability with an extension of a previously published method for >1,000 diseases, and compared the estimates to SNP-based heritability estimates from GWAS.

I have only few comments below:

1. Could the authors give the definition of the notations in formula (2) to (4)?
2. I am also confused by formula (4). It seems that $p(OS)$ is estimated as Nos/Nss , which is inherited from Lakhani et al. Nature Genetics 2019 (method formula 19). However, in the references that were cited in Lakhani et al., including [1] Neale. Twin. Res. 2003 [2] Scarr-Salapatek. Science 1971 [3] Beben Benyamin et al. Behavior Genetics 2005

$p(MZ)$ was proposed to be estimated as follows:

"[...] the proportion of MZ twin pairs was estimated as (Scarr-Salapatek, 1971): $1-2(\text{proportion of OS twin pairs})$ " (Neale. Twin. Res. 2003)

This description goes back to Scarr-Salapatek Science 1971 and also appeared in Beben Benyamin et al. Behavior

Genetics 2005.

I would interpret "proportion of OS twin pairs" as Nos/Nall, instead of Nos/Nss. I also verified this interpretation numerically based on the example in Beben Benyamin et al. Behavior Genetics 2005 Appendix Table 1, that Nos/Nall is how Beben Benyamin et al. calculated $p(MZ)$.

It also seems that from the Github code, the $p(MZ)$ is estimated as $1 - 2 * (Nos/Nall)$ in `h2_helpers.R` (line 102):

```
p_mz <- 1 - (2 * mf / (mm + ff + mf))
```

As this is part of the foundation of all estimates in the paper, could the authors clarify on this point and all potential all corresponding estimates?

3. The impact of these results are not clear to me. It is informative to have another set of narrow-sense heritability estimates, and the authors compared their results to previously published MaTCH and CaTCH study. However, what new insights we can gain from this study regarding the genetic basis of the diseases investigated, compared to MaTCH and CaTCH study, is not clear to me. The discussion mostly focus on the robustness and power of the heritability estimates. It would be great to also have some insight on the genetic finding itself.

(Remarks on code availability)

The code is clear and properly commented.

Reviewer #3

(Remarks to the Author)

This study leverages the Danish registries combining the Danish Civil Registration System and the National Patient Register to estimate the heritability of diseases in twins and siblings.

They apply a method described by Lakhani et al (referred to as CaTCH method) to estimate heritability from phenotypic correlations among twins where sex is known but information on zygosity is lacking. Subsequently, they include opposite-sex siblings in addition to opposite-sex twins to gain additional power.

Of the 1,146 traits that were analyzed, 598 had non-zero heritabilities. Left-truncation appeared to have a minimal effect on h^2 estimates when the cohort was split into the 1955 and 1977 birth cohort. Ultimately, the reported heritability estimates are compared to SNP-based h^2 estimates from the iPSYCH cohort using LDpred2.

The authors use an elegant way to estimate heritability in a unique and powerful cohort. Nevertheless, some sources of bias remain and could be assessed in more detail. Furthermore, the grouping of diseases in phecodes and relative low concordance previously published h^2 estimates illustrates that interpreting h^2 estimates remains challenging.

Main remarks:

As the authors state,

sex imbalance can bias estimates using the CaTCH method. The h^2 estimates might have been further inflated by grouping diseases in phecodes and including opposite sex sibs only. To illustrate, in the extreme example of the "Congenital anomalies of genital organs" phecode, the prevalence is similar in both sexes (ratio = 1.02). However, for all diseases within this phecode (e.g. "undescended testis", "doubling of uterus", etc.) opposite sex correlation is obviously 0. Therefore, it seems that any observed same sex correlation will result in a biased heritability estimate. For other diseases this effect might be more subtle.

Is it possible to assess this bias by including a cohort of same-sex siblings? Phecodes with a high correlation between same sex siblings and low correlation among opposite sex siblings (matched for age difference etc.) are susceptible to inflated CaTCH h^2 estimates due to sex bias of diseases within this phecode. Can this be used to filter out these phecodes (similar to how sex ratio is now used)? Additionally, is there evidence that this bias contributes to the difference between twin and sib-based estimates, given that the sib-based estimates only include additional opposite-sex sibs and not same-sex sibs?

The authors state the difference in the twin- h^2 and sib- h^2 estimates can indicate the effect of shared environment. Yet, this remains relatively unexplored. One could hypothesize the age difference between sibs affects the correlation between sibs (e.g. large age difference \rightarrow lower correlation within pairs). Is that indeed the case for some phecodes, and does this contribute to the difference between sib and twin based h^2 estimates?

In the comparison with CaTCH and MaTCH h^2 estimates which Danish estimates were used, those from the twin or sib cohort?

The authors fit a linear regression through the h^2 estimates from the Danish cohort and CaTCH. The correlation h^2 estimates for the phecodes seems low. However, it is hard to appreciate because standard errors for the h^2 estimates are not plotted, maybe the outliers are just less precise estimates. Will fitting a weighted regression that considers the standard error of h^2 estimates in both studies (IVW or similar) improve the model fit and yield a slope closer to 1? And does the model fit change when using the Danish twin or sib estimates?

A weighted regression could also be used when comparing the sib vs. twin-based estimates in figure 2.

In the comparison with the iPSYCH SNP-based h^2 estimates, is there any trend to observe which parameters explain the difference between twin-based h^2 estimates. Is it possible to obtain reliable estimates for polygenicity (P) from LDpred2 or alternatively use GCTB to estimate the number of non-zero SNPs and see signatures of purifying selection among common variants (S) that suggest these diseases are shaped by rare variants (using the GCTB summary-Bayes S/R methods). Or is the iPSYCH sample size too small for reliable estimates?

Minor remarks:

In table 1, the number of SS and OS twin pairs in the 1955 cohort do not add up to All pairs.

In table 1, opposite-sex pairs in the sibling cohorts (both 1977 and 1955) are not balanced between males and females. I would expect 50% males given that each pair includes a male and a female (similar to opposite-sex twin pairs). Can the authors explain this?

Equation 4 states that $p(OS) = N_OS / N_SS$, shouldn't this be $p(OS) = N_OS / N_all$. This might also be a typo in the original CaTCH paper?

Either way, when I plug in the numbers from table 1, $p(MZ|SS)$ values are slightly different to those reported. For example, in the 1977 cohort:

$N_OS = 15380$

$N_SS = 27330$

$N_total = 42710$

$pMZ = 1 - 2 * (N_OS / N_total) = 0.279794$

$pSS = N_SS / N_total = 0.639897$

$p = p(MZ|SS) = pMZ / pSS = 0.4372484$

Whereas the manuscript states that $p(MZ|SS) = 0.423$

Can the author clarify this, or point out a mistake in my calculation?

(Remarks on code availability)

Version 1:

Reviewer comments:

Reviewer #1

(Remarks to the Author)

I am satisfied with the responses, even though the authors essentially deflected my questions.

(Remarks on code availability)

Reviewer #2

(Remarks to the Author)

The authors addressed my comment on the equations, checked that the estimates in the manuscripts are correct, and revised them in the manuscript accordingly. The discussion on the heritability estimates is however still focused on the comparison between the methodologies and analyses between the current and previous studies. I was hoping the authors could also discuss the learnings from the current study on the genetic basis across phenotypes as described by the heritability estimates, and how does it compare to other similar estimates. But it seems that this is not the focus of the current study or outside of the scope of the current study. I have no further comments.

(Remarks on code availability)

I have not re-review the codes.

Reviewer #3

(Remarks to the Author)

I thank the authors for their careful response and revisions. Remaining questions are discussed in the Discussion and can indeed be a topic of future research.

(Remarks on code availability)

I have not reviewed the code in details, but looked up some functions and calculations. It was well documented and clearly written.

Letter of reply

First, we would like to thank the reviewers for taking the time to read and provide such detailed and concise comments to our manuscript. We greatly appreciate your efforts and feel the additions and revisions have improved the manuscript.

Please find our replies (in blue) and the revised text (in black, additions underlined) below. Edits in the manuscript are marked in red.

Reviewer comments

Reviewer #1 (Remarks to the Author)

The study is using information about same-sex and distinct-sex siblings plus share or not shared date of birth of siblings as a proxy to monozygotic and dizygotic twin analysis.

The study is valuable and seems to agree well with previously published data.

Response

We thank Reviewer #1 for noting the value of our study and its agreement with previous findings.

This reviewer has the following questions (ideally answered in the revised manuscript).

Question 1

Since mother and father information is available, why the authors did not consider analyzing heritability based on whole family rather on siblings?

Response

Thank you for raising this point. It is indeed possible to incorporate pedigree-based methods using parent–offspring and more distant relatives, thereby enabling estimates that differentiate additive genetic and shared environmental variance. In fact, recent work on Danish registry data (e.g. Westergaard et al., Nature Communications 2024) has successfully implemented extended-pedigree heritability estimation in Danish cohorts, illustrating the feasibility and added insight of such approaches.

However, by restricting to siblings we hope to minimize the risk of shared environmental effects confounding the heritability estimate. Also, our study's goal was to evaluate the scalability and applicability of twin- and sibling-based correlation methods across a large number of phenotypes. Adding full pedigree-based analyses would constitute a substantial methodological expansion, shifting the focus and structure of the paper. A comprehensive pedigree-based analysis is therefore beyond the current scope, but we agree that this represents a valuable avenue for future work and have expanded the Discussion section as described below.

Discussion section:

P. 29: To address this, future work could incorporate traditional zygosity-based estimates from the Danish Twin Registry. In the present study, we chose to focus on twins and siblings rather than full pedigrees to reduce the risk of shared environmental factors confounding the heritability estimates, but the framework could also be expanded with pedigree-based methods, for which the Danish registries are ideal [16, 45]. Incorporating these methods would also allow for further validation of our results.

Question 2

Did the authors consider the issues of statistical efficiency? The equations they used were developed in pre-computer era for ease of manual computation.

Response

Thank you for raising this point. Although the correlation-based equations we use originate from classical twin modelling, they remain sufficiently efficient and with modern computational resources, their implementation is straightforward. In practice, the estimation is computationally light: for each phenotype, model fitting requires less than a minute in the (smaller) twin cohorts and up to a few minutes in the (very large) sibling cohorts on a standard workstation, making it feasible to apply these methods across more than one thousand phenotypes.

Question 3

There were a number of published studies regarding estimating heritability from administrative records the authors seem to be unaware of. Would the author consider augmenting their comparisons to prior estimates? (Remarks on code availability)

Response

We thank the reviewer for highlighting this. Several large-scale studies have indeed estimated heritability using administrative data [REFs 13-16 in the paper] and we have added their references as described below. We do not compare directly to these because they and the Danish pedigree study (Westergaard et al., 2024) apply a different methodological framework. We now highlight these points more clearly in the Introduction as described below.

Introduction section

P. 4: However, the reliance on insurance-based data in the CaTCH study introduces several well-recognised limitations, including ascertainment bias, limited follow-up, and incomplete phenotype capture [13-16].

Reviewer #2 (Remarks to the Author)

To estimate narrow-sense heritability, the authors identified all twin and sib-pairs born in Denmark between 1955 and 2021 using the Danish Civil Registration System, and created a twin cohort and a sibling cohort that is inclusive of the twin cohort. The authors performed data linkage to the Danish National Patient Register was done for both cohorts to identify disease diagnostic records. The authors then estimated narrow-sense heritability with an extension of a previously published method for >1,000 diseases, and compared the estimates to SNP-based heritability estimates from GWAS.

Response

We thank Reviewer #2 for the constructive summary of our methods and analyses.

I have only few comments below:

Question 1

Could the authors give the definition of the notations in formula (2) to (4)?

Response

We thank the reviewer for pointing out the need to define the notations more explicitly. In the revised manuscript, we have added explicit definitions of all notations used, to ensure clarity and consistency. The additions have been made as described below.

Methods section

P. 8:

$$h^2 = \frac{2}{p}(r_{SS} - r_{OS}) \quad (21)$$

where r_{SS} and r_{OS} are the same-sex and opposite-sex correlations respectively, and p is the probability of a same-sex pair being monozygotic. Consequently, twin pairs were stratified into same-sex and opposite-sex twins, where opposite-sex pairs will necessarily be dizygotic and same-sex pairs will be a mixture of monozygotic and dizygotic twins.

P. 8:

$$p = p(MZ|SS) = \frac{p(MZ)}{p(SS)} \quad (32)$$

here, $p(MZ|SS)$ denotes the probability that a twin pair is monozygotic, given that they are same-sex. This is calculated as the ratio of the overall probability of being monozygotic, $p(MZ)$ to the probability of being a same-sex pair, $p(SS)$.

The overall probability of being monozygotic, $p(MZ)$, can be expressed in terms of the proportion of opposite-sex pairs:

$$p(MZ) = 1 - 2p(OS) = 1 - 2\frac{N_{OS}}{N_{all}} \quad (34)$$

here, $p(OS)$ is the probability of being an opposite-sex pair, with N_{OS} being the number of opposite-sex pairs and N_{all} being the total number of pairs.

Finally, the probability of being a same-sex pair is given by:

$$p(SS) = \frac{N_{SS}}{N_{all}} \quad (54)$$

where N_{SS} is the number of same-sex pairs.

Question 2

I am also confused by formula (4). It seems that $p(OS)$ is estimated as N_{OS}/N_{SS} , which is inherited from Lakhani et al. Nature Genetics 2019 (method formula 19). However, in the references that were cited in Lakhani et al., including

[1] Neale. Twin. Res. 2003

[2] Scarr-Salapatek. Science 1971

[3] Beben Benyamin et al. Behavior Genetics 2005

$p(MZ)$ was proposed to be estimated as follows:

“[...] the proportion of MZ twin pairs was estimated as (Scarr-Salapatek, 1971): $1-2(\text{proportion of OS twin pairs})$ ” (Neale. Twin. Res. 2003)

This description goes back to Scarr-Salapatek Science 1971 and also appeared in Beben Benyamin et al. Behavior Genetics 2005.

I would interpret “proportion of OS twin pairs” as N_{OS}/N_{all} , instead of N_{OS}/N_{SS} . I also verified this interpretation numerically based on the example in Beben Benyamin et al. Behavior Genetics 2005 Appendix Table 1, that N_{OS}/N_{all} is how Beben Benyamin et al. calculated $p(MZ)$.

It also seems that from the Github code, the $p(MZ)$ is estimated as $1 - 2 * (N_{OS}/N_{all})$ in `h2_helpers.R` (line 102):

```
p_mz <- 1 - (2 * mf / (mm + ff + mf))
```

As this is part of the foundation of all estimates in the paper, could the authors clarify on this point and all potential all corresponding estimates?

Response

We thank the reviewer for identifying this mistake! The reviewer is correct that the expression for $p(OS)$ in Equation 4 was written incorrectly in the original manuscript. As also noted by Reviewer #3, the correct definition is $p(OS) = N_{OS} / N_{all}$, consistent with the formulations in the cited literature (e.g. Scarr-Salapatek 1971; Neale 2003; Benyamin et al. 2005) and this was correctly implemented in our (publicly available) code. Hence, this was a typographical error in the presentation of the formula only. All estimates reported in the paper were obtained using the correct expression and are therefore unaffected. We have corrected this in the revised manuscript as marked in red below.

Methods section:

$$\mathbf{P. 8: } p(MZ) = 1 - 2p(OS) = 1 - 2 \frac{N_{OS}}{N_{all}} \quad (45)$$

Question 3

The impact of these results are not clear to me. It is informative to have another set of narrow-sense heritability estimates, and the authors compared their results to previously published MaTCH and CaTCH study. However, what new insights we can gain from this study regarding the genetic basis

of the diseases investigated, compared to MaTCH and CaTCH study, is not clear to me. The discussion mostly focus on the robustness and power of the heritability estimates. It would be great to also have some insight on the genetic finding itself.

Response

We agree that the motivation and impact of the study could be improved. Our study complements previous work by applying the CaTCH framework to Danish nationwide health registers, which differ fundamentally from the insurance-based data used in CaTCH. E.g., by having a near-complete population coverage and long follow-up, our study is less susceptible to selection bias (e.g. ascertainment bias). We also examine the impact of calendar time on the heritability estimates by comparing two birth cohorts, one with individuals born from 1977 on onwards and on from 1955 and onwards. The former has complete follow-up on all individuals from they are born until the end of 2021, while the latter goes as far back as linkage of siblings was possible, which widened the age distribution of the individuals available, yet left truncated the data. We found that left truncation had little effect on the heritability estimates. Moreover, we systematically compare twin- and sibling-based estimates and identify patterns across disease domains that suggest differing contributions of shared environmental factors. Finally, restricting to psychiatric and neurological disorders, we compare these estimates with SNP-heritability estimates providing a broader perspective on genetic architecture of these at the population level. We have revised the Introduction and Discussion to make these contributions clearer.

Introduction section

P. 4: Each approach is subject to distinct assumptions and artefacts, complicating cross-study interpretation and limiting the utility of these estimates for translational research. Large-scale meta-analyses such as the Meta-Analysis of Twin Correlations and Heritability (MaTCH) have sought to provide systematic overviews of heritability across traits by aggregating results from thousands of classical twin studies [3]. While these efforts have been invaluable as reference resources, they necessarily combine heterogeneous cohorts with differing recruitment strategies, diagnostic practices, and follow-up durations, which can introduce variability and limit comparability across disease domains and populations.

P. 4: However, the reliance on insurance-based data in the CaTCH study introduces several well-recognised limitations, including ascertainment bias, limited follow-up, and incomplete

phenotype capture [13-16]. In contrast, the Danish national health registries provide nationwide, population-based coverage with virtually complete follow-up and systematic recording of all hospital contacts, thereby minimising selection and ascertainment biases. In this study, we therefore apply and extend the novel CaTCH framework to these registries, incorporating both twin and sibling data to estimate heritability for more than one thousand disease outcomes.

Reviewer #2 (Remarks on code availability)

The code is clear and properly commented.

Response

We thank Reviewer #2 for the positive assessment of the code availability and documentation.

Reviewer #3 (Remarks to the Author)

This study leverages the Danish registries combining the Danish Civil Registration System and the National Patient Register to estimate the heritability of diseases in twins and siblings. They apply a method described by Lakhani et al (referred to as CaTCH method) to estimate heritability from phenotypic correlations among twins where sex is known but information on zygosity is lacking. Subsequently, they include opposite-sex siblings in addition to opposite-sex twins to gain additional power.

Of the 1,146 traits that were analyzed, 598 had non-zero heritabilities. Left-truncation appeared to have a minimal effect on h^2 estimates when the cohort was split into the 1955 and 1977 birth cohort. Ultimately, the reported heritability estimates are compared to SNP-based h^2 estimates from the iPSYCH cohort using LDpred2.

The authors use an elegant way to estimate heritability in a unique and powerful cohort. Nevertheless, some sources of bias remain and could be assessed in more detail. Furthermore, the grouping of diseases in phecodes and relative low concordance previously published h^2 estimates illustrates that interpreting h^2 estimates remains challenging.

Response

We thank Reviewer #3 for the careful summary and thoughtful assessment of our study. We acknowledge that several sources of bias remain, as highlighted in the reviewer's comments. We address these points in detail throughout our point-by-point responses below.

Main remark 1

As the authors state, sex imbalance can bias estimates using the CaTCH method. The h^2 estimates might have been further inflated by grouping diseases in phecodes and including opposite sex sibs only. To illustrate, in the extreme example of the "Congenital anomalies of genital organs" phecode, the prevalence is similar in both sexes (ratio = 1.02). However, for all diseases within this phecode (e.g. "undescended testis", "doubling of uterus", etc.) opposite sex correlation is obviously 0. Therefore, it seems that any observed same sex correlation will result in a biased heritability estimate. For other diseases this effect might be more subtle.

Response

We thank the reviewer for this important point. As you correctly point out, we perform our analyses by dividing all twin pairs into same-sex and opposite-sex pairs, with all full sibling pairs, same-sex as well as opposite-sex pairs being added to the opposite-sex pair category, in our sibling analyses. This complicates the estimation in sex-specific conditions and, as you astutely note, within aggregated phecodes (e.g., undescended testis, uterine anomalies) where opposite-sex correlations are zero. In the revised manuscript, we have therefore decided to expand our filtering by also removing any parent-phecode of phecodes categorised as imbalanced by our own definition, which resulted in 3-7 phecodes being removed per birth cohort. We have adapted the Method section to show this and updated figures as well as Table 2 to reflect the new counts.

Methods section

P. 7: Similarly, phecode categories with a female-to-male or male-to-female ratio of more than 5 were filtered away (Table 2) along with their parental phecodes to avoid categories that appear to have a balanced sex distribution, but are in fact a mixture of two or more imbalanced categories (e.g., *Congenital anomalies of female genital organs (751.11)*, *Congenital anomalies of male genital organs (751.12)* which aggregate in the category *Congenital anomalies of genital organs (751.1)*) [2].

P. 9: In the analyses with the full sibling cohort, all non-twin sibling pairs in the cohort, irrespective of sex composition, were categorised as opposite-sex pairs, reflecting how full siblings are as genetically similar as dizygotic twins.

Results section

P. 13: Table 1. Summary of twin and sibling cohorts. Any counts are currently rounded to nearest 10. *Including twins. **Including both same-sex and opposite-sex sibling pairs along with the opposite-sex twin pairs.

Discussion section

P. 28: However, the method does not apply to sex-specific phenotypes which restricts our ability to assess the heritability of conditions that predominantly or exclusively affect one sex, such as *placenta previa* or *prostate cancer*, or phecodes that are groupings of sex-specific subgroups, such as *Congenital anomalies of genital organs* that contain the subgroups *Congenital anomalies of female genital organs* and *Congenital anomalies of male genital organs*.

Main remark 2

Is it possible to assess this bias by including a cohort of same-sex siblings? Phecodes with a high correlation between same sex siblings and low correlation among opposite sex siblings (matched for age difference etc.) are susceptible to inflated CaTCH h2 estimates due to sex bias of diseases within this phecode. Can this be used to filter out these phecodes (similar to how sex ratio is now used)? Additionally, is there evidence that this bias contributes to the difference between twin and sib-based estimates, given that the sib-based estimates only include additional opposite-sex sibs and not same-sex sibs?

Response

We thank the reviewer for this useful suggestion. We agree that comparing same-sex and opposite-sex sibling correlations could help identify phecodes where sex imbalance may bias heritability estimates. However, we decided to instead filter out any parent phecode of the phecodes that we identified as imbalanced regarding the sex distribution, as described in the response to Main Remark 1 above.

Main remark 3

The authors state the difference in the twin-h2 and sib-h2 estimates can indicate the effect of shared environment. Yet, this remains relatively unexplored. One could hypothesize the age difference between sibs affects the correlation between sibs (e.g. large age difference -> lower correlation within pairs). Is that indeed the case for some phecodes, and does this contribute to the difference between sib and twin based h2 estimates?

Response

We thank the reviewer for this valuable suggestion. We agree that age differences between siblings could influence their phenotypic correlations, which is an important consideration. Although this would be a highly interesting aspect to explore, the methodological work required to account for within-family age differences is beyond the scope of the present study. We hope to investigate this question in future work.

Discussion

P. 29: Furthermore, we did not account for age differences within sibling pairs. Larger age gaps may reduce shared environmental exposure and affect within-family phenotypic correlations, which could partly contribute to the observed differences between sibling- and twin-based heritability estimates. Although the average age difference in our cohort was small (2.4 years), future work could examine this more directly by adjusting for age spacing in the linear mixed model.

Main remark 4

In the comparison with CaTCH and MaTCH h2 estimates which Danish estimates were used, those from the twin or sib cohort?

Response

Thank you for highlighting this ambiguity. In the comparison with CaTCH and MaTCH, we used our twin-based estimates, as both CaTCH and MaTCH are based on twin cohorts. We have clarified this point in the revised manuscript as described below.

Methods section

P. 11: We mapped phecodes of our twin estimates to the corresponding phecodes of the CaTCH study.

Results section

P. 22: To evaluate our findings, we examined the degree of concordance with prior literature by comparing our twin heritability estimates to the original CaTCH estimates as well as those of the Meta-Analysis of Twin Correlations and Heritability (MaTCH) [3] (Figure 3).

P. 22: Our results were largely consistent with those of the MaTCH study (Figures 3a and c), but our mean twin heritability of 0.453 (95% CI: 0.435, 0.470) was smaller than their mean of $h^2 = 0.593$ (95% CI: 0.577, 0.608).

P. 23: Figure 3. Comparison of Danish twin heritability estimates with previous twin-based studies.

Discussion section

P. 27: Further comparisons of our twin heritability estimates with the twin estimates from CaTCH and MaTCH studies revealed that while our results were generally larger than those reported in CaTCH, they were closely aligned with those in MaTCH, particularly for neurological, psychiatric, and musculoskeletal disorders.

Main remark 5

The authors fit a linear regression through the h^2 estimates from the Danish cohort and CaTCH. The correlation h^2 estimates for the phecodes seems low. However, it is hard to appreciate because standard errors for the h^2 estimates are not plotted, maybe the outliers are just less precise estimates. Will

fitting a weighted regression that considers the standard error of h^2 estimates in both studies (IVW or similar) improve the model fit and yield a slope closer to 1? And does the model fit change when using the Danish twin or sib estimates?

Response

Thank you, this is a very good point. We have applied an inverse-variance weighted regression to figure 3b + c and added error bars to all estimates. The inverse-variance weighting improves the fit in both regressions as both intercepts (α) decreased towards zero, and both slopes (β) increased, becoming closer to 1. We have expanded the figure legend to make this clear as described below.

We thought it was a very interesting idea to attempt to fit the same model with our sibling estimates, however, this did not improve the regression fits, but the resulting plots have been added as Supplementary figure 6.

Results section

P. 23: (b) Weighted regression comparing Danish heritability estimates to those from the CaTCH study. Each point represents a phenotype with non-zero heritability in both datasets with error bars. The red line indicates the weighted linear regression fit, while the dashed black line represents the identity line (slope = 1). (c) Weighted regression comparing Danish estimates with those from the MaTCH meta-analysis.

Main remark 6

A weighted regression could also be used when comparing the sib vs. twin-based estimates in figure 2.

Response

We agree that a weighted regression is a more appropriate approach when comparing the sibling- and twin-based estimates. In fact, this was already implemented in the analysis shown in Figure 2, although this was not made explicit in the original version. We have now clarified this in the Results section as described below.

Results section

P. 21: Comparison of liability-scale heritability estimates derived from sibling pairs (x-axis) with those from twin pairs (y-axis) using (a) the 1955 cohort and (b) the 1977 cohort. Each point represents a phenotype, coloured according to its

functional domain and sized by weight. The red line indicates the weighted linear regression fit, while the dashed black line represents the identity line (slope = 1).

Main remark 7

In the comparison with the iPSYCH SNP-based h^2 estimates, is there any trend to observe which parameters explain the difference between twin-based h^2 estimates. Is it possible to obtain reliable estimates for polygenicity (P) from LDpred2 or alternatively use GCTB to estimate the number of non-zero SNPs and see signatures of purifying selection among common variants (S) that suggest these diseases are shaped by rare variants (using the GCTB summary-Bayes S/R methods). Or is the iPSYCH sample size too small for reliable estimates?

Response

We thank the reviewer for this insightful suggestion. Following the recommendation, we used LDpred2-auto to estimate the degree of polygenicity (P) and the selection coefficient (α) for each iPSYCH-based phenotype. These results are presented in Supplementary Figure 7, and we have expanded the results section as described below.

Methods section

P. 12: The resulting summary statistics were used to compute the h_{SNP}^2 , polygenicity (p) and selection coefficient (α) using the Bayesian framework LDpred2-auto [35, 36].

Results section

P. 24: To further characterise these patterns, we examined the estimated polygenicity (P) and selection coefficients (α) for each disorder (Supplementary Figure 7). Psychiatric disorders exhibited higher polygenicity and more negative selection coefficients than neurological disorders, indicating that their genetic architecture is shaped by a larger number of small-effect variants subject to stronger purifying selection.

Minor remark 1

In table 1, the number of SS and OS twin pairs in the 1955 cohort do not add up to All pairs.

Response

We thank Reviewer #3 for this observation. To comply with GDPR requirements, the numbers in Table 1 have been rounded, which explains why the sum of same-sex and opposite-sex pairs does not exactly equal the reported total. The exact counts will be made available upon final publication of the article.

Minor remark 2

In table 1, opposite-sex pairs in the sibling cohorts (both 1977 and 1955) are not balanced between males and females. I would expect 50% males given that each pair includes a male and a female (similar to opposite-sex twin pairs). Can the authors explain this?

Response

As you correctly point out, there is not an equal distribution of sexes in the OS group in the sibling cohorts. This is due to the fact that all non-twin sibling pairs are classified into the OS group regardless of their actual sex configuration. We have expanded the description of Table 1, as described below and in the response to Main Remark 2, to clarify this.

Our rationale for this is based on the model, which assumes that all OS twin pairs will necessarily be dizygotic, while SS pairs will be a mixture of monozygotic and dizygotic. Since non-twin siblings on average share 50% of their segregating DNA, just like dizygotic twin pairs, it is consistent with the model to include them in this group.

Results section

P. 13: Table 1. Summary of twin and sibling cohorts. Any counts are currently rounded to nearest 10. *Including twins. **Includes both same-sex and opposite-sex sibling pairs along with the opposite-sex twin pairs.

Minor remark 3

Equation 4 states that $p(OS) = N_OS / N_SS$, shouldn't this be $p(OS) = N_OS / N_all$. This might also be a typo in the original CaTCH paper?

Response

We thank the reviewer for carefully noting this issue. The reviewer is absolutely correct that the expression shown in Equation 4 was a typographical error and should read $p(OS) = N_OS / N_all$.

We have corrected this in the revised manuscript and note that this was a presentation error only; all reported estimates are based on the correct definition as implemented in the code. Thank you for pointing this out, the edit made to Equation 4, is marked in red below.

Methods section:

$$\mathbf{P. 8: } p(MZ) = 1 - 2p(OS) = 1 - 2 \frac{N_{OS}}{N_{all}} \quad (6)$$

Minor remark 4

Either way, when I plug in the numbers from table 1, $p(MZ|SS)$ values are slightly different to those reported. For example, in the 1977 cohort:

$$N_OS = 15380$$

$$N_SS = 27330$$

$$N_total = 42710$$

$$pMZ = 1 - 2 * (N_OS / N_total) = 0.279794$$

$$pSS = N_SS / N_total = 0.639897$$

$$p = p(MZ|SS) = pMZ / pSS = 0.4372484$$

Whereas the manuscript stats that $p(MZ|SS) = 0.423$

Can the author clarify this, or point out a mistake in my calculation? (Remarks on code availability)

Response

We thank the reviewer for carefully checking the derivation. The calculation is indeed correct and reflects the expected value based on the updated cohort numbers used in our analysis. The discrepancy arises because the estimate reported in the manuscript had not been updated following our extension of the cohort follow-up from 2018 to 2021. We have now corrected the manuscript accordingly to reflect the updated value of $p(\text{MZ}|\text{SS}) = 0.437$. This was a presentation error only; all results and code use the correct and current numbers.